# Gut–Kidney Axis on Chip for Studying Effects of Antibiotics on Risk of Hemolytic Uremic Syndrome by Shiga Toxin-Producing *Escherichia coli*

**DOI:** 10.3390/toxins13110775

**Published:** 2021-11-02

**Authors:** Yugyeong Lee, Min-Hyeok Kim, David Rodrigues Alves, Sejoong Kim, Luke P. Lee, Jong Hwan Sung, Sungsu Park

**Affiliations:** 1Department of Biomedical Engineering, Sungkyunkwan University (SKKU), Suwon 16419, Korea; gyeong260@skku.edu; 2School of Mechanical Engineering, Sungkyunkwan University (SKKU), Suwon 16419, Korea; mhkim967@g.skku.edu (M.-H.K.); david.r.alves@tecnico.ulisboa.pt (D.R.A.); 3Department of Chemical Engineering, Hongik University, Seoul 04066, Korea; 4Department of Bioengineering, Instituto Superior Técnico, Universidade de Lisboa, 1362035 Lisboa, Portugal; 5Department of Internal Medicine, Seoul National University Bundang Hospital, Seongnam 13620, Korea; sejoong@snubh.org; 6Institute of Quantum Biophysics (IQB), Department of Biophysics, Sungkyunkwan University (SKKU), Suwon 16419, Korea; LPLEE@bwh.harvard.edu; 7Renal Division, Department of Medicine, Brigham and Women’s Hospital, Harvard Medical School, Boston, MA 02115, USA; 8Division of Engineering in Medicine, Department of Medicine, Brigham and Women’s Hospital, Harvard Medical School, Boston, MA 02115, USA

**Keywords:** hemolytic–uremic syndrome (HUS), multi-organ-on-a-chip, *Escherichia coli* infection, antibiotics, Shiga toxin

## Abstract

Shiga toxin-producing *Escherichia coli* (STEC) infects humans by colonizing the large intestine, and causes kidney damage by secreting Shiga toxins (Stxs). The increased secretion of Shiga toxin 2 (Stx2) by some antibiotics, such as ciprofloxacin (CIP), increases the risk of hemolytic–uremic syndrome (HUS), which can be life-threatening. However, previous studies evaluating this relationship have been conflicting, owing to the low frequency of EHEC infection, very small number of patients, and lack of an appropriate animal model. In this study, we developed gut–kidney axis (GKA) on chip for co-culturing gut (Caco-2) and kidney (HKC-8) cells, and observed both STEC O157:H7 (O157) infection and Stx intoxication in the gut and kidney cells on the chip, respectively. Without any antibiotic treatment, O157 killed both gut and kidney cells in GKA on the chip. CIP treatment reduced O157 infection in the gut cells, but increased Stx2-induced damage in the kidney cells, whereas the gentamycin treatment reduced both O157 infection in the gut cells and Stx2-induced damage in the kidney cells. This is the first report to recapitulate a clinically relevant situation, i.e., that CIP treatment causes more damage than gentamicin treatment. These results suggest that GKA on chip is very useful for simultaneous observation of O157 infections and Stx2 poisoning in gut and kidney cells, making it suitable for studying the effects of antibiotics on the risk of HUS.

## 1. Introduction

Shiga toxin-producing *Escherichia coli* (STEC), including the serotype O157:H7 (O157), infects the human gastrointestinal tract, potentially leading to the development of hemolytic–uremic syndrome (HUS), a syndrome characterized by mechanical hemolytic anemia, thrombocytopenia, and kidney dysfunction [1]. HUS is life-threatening. In 2017, STEC infections were reported in a small community near the Arizona–Utah border, and two of 12 children died [2]. In 2019, there was an outbreak of O157 in Brazil, and three out of 24 patients developed HUS, with one death [3]. This high mortality is partly owing to the lack of proper antibiotic treatments for patients with STEC. Antibiotics inhibit the growth of STEC, but quinolone antibiotics, including ciprofloxacin (CIP), can increase the production of Shiga toxin 2 (Stx2) by increasing the expression of the stx2 gene. This is a major contributor to the development of HUS [4,5]. Stx2 consists of one A subunit and five B subunits. The pentamer of the B subunit binds to the cellular receptor globotriaosylceramide-3 (Gb3) [6], which is found in kidney epithelial tissues [7,8] and the central nervous system [7], causing kidney tubular injury and kidney failure, and leading to the development of HUS [4]. In detail, Stx2 released from STEC passes through the gut cell layer, circulates in the body along the bloodstream, and enters the kidney via binding to Gb3 receptors [8]. For the treatment of HUS, only supportive treatments such as intravenous fluid infusion, blood transfusions, and kidney dialysis are available [5]. Despite the urgent need for studies on HUS development after antibiotic treatment, such studies have been very limited, owing to the low periodicity of its cases and lack of the required number of patients required [9]. Moreover, corresponding animal models are not well-established; this is because the toxicity of Stx in animal models differs from that in humans, owing to differences in Gb3 expression and localization between species [9]. Therefore, it is necessary to develop an in vitro STEC model to develop an antibiotic therapy that avoids damage to kidney cells.

Organ on chip (OOC) is a microphysiological system that mimics the functions of human organs [10,11,12]. It consists of microchannels that allow cells to grow in three dimensionally grown through perfusion. Although the human body comprises many organs and the interactions between different organs play important roles in maintaining homeostasis and disease progression, OOC has generally been limited to simulating a single organ. To overcome this limitation, multi-organ-on-a-chip (MOOC) has been developed to simulate whole-body physiologies and responses to drugs. MOOC can be constructed on a single chip [13] or on multiple separate chips connected to fluidic tubes [14]. The main advantage of the MOOC platform is that it allows for the realization of organ-organ interactions [15]. The complex processes that drugs undergo inside the body, including their absorption, distribution, metabolism, and excretion (ADME), involve different organs in the body, and earlier proof-of-concept developments of MOOC for predicting the pharmacokinetics–pharmacodynamics of drugs have been reported [16,17,18]. Therefore, MOOC has made it possible to elucidate disease mechanisms and/or develop therapeutic strategies for diseases [18]. In particular, the pivotal roles played by gut microbiota in the human body have been a major recent research interest, but only a few OOC models for reproducing the gut microbe co-culture have been reported, probably owing to technical difficulties in providing adequate culture environments for both the gut cells and microbes [19].

In this study, gut–kidney axis (GKA) on chip for the co-culturing of gut and kidney cells was developed to simulate HUS, i.e., an STEC infection occurring in the gut, and the damage to the kidneys caused by Stx2 released from the infected gut (Figure 1a). Gut (Caco-2) and kidney (HKC-8) cells were first cultured on the main body and module, respectively, until they formed monolayers, and were then co-cultured by inserting the module into the chip (Figure 1b). Cells in the chip were perfused with media at a constant flow rate using a tilting machine for 10 days. The distribution of Stx2 inside the chip was calculated by simulation, while the effects of the toxin on the gut and kidney cells were studied by measuring the cell viability and cell–cell junction integrity. To investigate the differential effects of antibiotics to STEC infections on kidney damage, the damage to the gut and kidney cells in the chip was assessed while treating O157-infected gut cells in the chip with either CIP or gentamicin (GEN). This is the first demonstration of the MOOC for examining the effect of antibiotics treatment to STEC infection on kidney damage.

## 2. Results and Discussions

### 2.1. Fabrication and Operation of GKA on Chip

Currently, the co-culturing of microbes in OOC models featuring a single gut has been well established [20]. However, many conditions in the human body develop and progress through cross-organ communication [21]. Hence, MOOC can provide new insights into in vitro cross-organ interactions that cannot be presented in single-organ OOC. In addition, the modular concept of MOOC, such as in our model, can offer convenience in analyzing each module, and allow for customized experimental protocols by changing the modules to those with special stimulation [22]. Our MOOC consists of two distinct parts: the gut module (main body) and kidney module (Figure 2a,b). This allows us to conveniently care for and analyze both modules since they can be easily assembled and disassembled when necessary. This also allows for the direct investigation of the effects of antibiotics or toxins on kidney cells through non-invasive analysis such as transepithelial electrical resistance (TEER) measurement. Gravity-induced perfusion through tilting the chip on the tilting machine washes wastes out of the cells, and thus the cells are maintained better and longer than in conventional well models (Figure 2c,d). Furthermore, this approach can improve the throughput of data by reducing labors of connecting external parts to perfuse each chip, and the chips are more easily cared for by the user.

### 2.2. Co-Culture of Caco-2 and HKC-8 Cells on GKA on Chip

As the cell culture media for Caco-2 and HKC-8 cells are different, there was a concern that this difference might have detrimental effects on each cell line when they were co-cultured on the chip. To measure the viability of each cell, LIVE/DEAD staining and an EZ-CytoX assay of Caco-2 and HKC-8 cells, either mono-cultured or co-cultured for 3 days, were conducted. Our results showed that there was no significant difference in the cell viability between the mono-culture of each cell line and the co-culture of both cell lines on the chip (Figure 3a,b). This was because HKC-8 cells were separately cultured in the module until they formed a monolayer, and were later co-cultured with Caco-2 cells by inserting the module into the chip. Our chip was able to co-culture two types of cells that are otherwise difficult to grow together (because of their different needs for the cell culture medium). If the need for the culture medium is not fulfilled in the early stages of cell growth, differentiation or maturation will be inhibited [23]. Our chip overcame this by culturing two types of cells independently in separable inserts until they reached a stable state, and them assembling them with the chip for the co-culture experiments.

The junctional integrity in a cell monolayer is crucial for the movement of molecules across the monolayer [24]. TEER is used as an indicator of the barrier formation in a cell monolayer [25]. Since it is difficult to measure TEER in a microfluidic device [26] and the gut module could not be separated from the chip, the junctional integrity of the gut module was evaluated by immunostaining the tight junction marker occludin [27] on day 7. The cell boundary was clearly observed, confirming the formation of an intact epithelial barrier by Caco-2 during co-culturing (Figure 3c). Since the kidney module was easily detachable from the chip, its junctional integrity was determined by monitoring the TEER values over 8 days. HKC-8 cells were mono-cultured in the kidney module for 4 days, and then were co-cultured in the chip by inserting the module into the chip. The TEER values in the kidney module increased until day 5 and then stabilized at 20 ohms × cm^2^, without further significant changes throughout the following days (Figure 3d). The TEER values were higher than previously reported values [28], indicating that the kidney cells in the chip formed tight barriers. Taken together, the results showed that the two cell lines could be co-cultured in the chip without compromising the barrier function or viability.

### 2.3. Effect of Stx2 on Viability and Barrier Integrity of Caco-2 and HKC-8 Cells

The viability of Caco-2 and HKC-8 cells was investigated when purified Stx2 was applied at a concentration of 3.03 to 21.2 nM in the gut module of GKA on chip for 72 h. As Stx2 concentration increased, the viability of Caco-2 cells did not decrease, but the viability of HKC-8 cells decreased. The inhibitory effect of Stx2 on HKC-8 cells was evident at 21.2 nM (*p* < 0.001) (Figure 4a). In contrast to HKC-8 cells, Caco-2 cells were quite resistant to Stx2 (*p* < 0.05 at 21.2 nM), probably owing to the lack of Gb3, consistent with previous reports [29,30].

Transport of Stx2 produced by O157 residing in the gut module inside the chip is crucial to the toxic response of HKC-8 cells in the kidney module. Simple mass balance Equations (2) and (3) for the two-module model were set up and used to simulate the transport of Stx2 from the gut to the kidney module. The simulation showed that the transport of Stx2 from the gut to the kidney module was limited, as only 0.12% of the initial concentration was delivered to the kidney module after 72 h (Figure 4b). The concentration of Stx2 in the gut module 72 h after treatment with 21.2 nM of Stx2 was predicted to be 20.8 nM, whereas the concentration of Stx2 in the kidney module was predicted to be 25.5 pM. Such a low concentration of Stx2 in the kidney module was sufficient to elicit a toxic response, as also observed in a 96-well plate (Appendix A).

To confirm the effect of Stx2 on the integrity of Caco-2 and HKC-8 cells, 21.2 nM of Stx2 was added to Caco-2 cells in Transwell, and the solution was treated in the gut module of GKA on chip for 72 h. TEER values of Caco-2 cells and HKC-8 cells were measured 72 h after the treatment (Figure 4c,d). In general, it is difficult to measure the TEER values in a microfluidic device [26], because the TEER electrodes are designed for Transwell, and are not fit for a chip. Since the gut module could not be separated from the chip (Figure 2a), we could only obtain TEER values from the kidney module. To indirectly estimate the effect of Stx2 on the integrity of Caco-2 cells, Caco-2 cells were cultured in Transwell under the same conditions as on the chip, and their TEER values were measured after Stx2 treatment. There was no significant difference in the TEER values between Caco-2 cells with and without Stx2 treatment at 21.2 nM for 72 h (Figure 4c), indicating that the cellular integrity of the gut cells was not affected by Stx treatment. HKC-8 cells showed a stable TEER value of approximately 20 ohms × cm^2^ on day 6 (Figure 4b). The treatment reduced the TEER value to 10 ohms × cm^2^ on day 6, i.e., significantly lower than the value without the treatment (Figure 4d). This result was consistent with the reduction in the viability of HKC-8 cells when treated with Stx2 (Figure 4a). It was reported that Stx2 from O157 infection caused damage to the kidney epithelial cells, reducing their cellular integrity [31]. Taken together, these results suggest that the small amounts of Stx2 produced by O157 in Caco-2 cells may be detrimental to HKC-8 cells in the chip.

### 2.4. Evaluating the Risk of HUS in the Kidney by CIP and GEN Using the Chip

To determine whether the antibiotics themselves were toxic to Caco-2 and HKC-8 cells, both types of cells were separately cultured in the 96-well plate, and were treated with either CIP or GEN. Neither significantly affected the viability of Caco-2 and HKC-8 cells (Appendix A), indicating that at their respective minimum inhibitory concentrations (MICs), they were not toxic to either cell type.

To demonstrate that O157 could infect Caco-2 cells and its toxins could cause cytotoxic effects in HKC-8 cells on the chip, LIVE/DEAD staining was performed 72 h after the gut modules were seeded with O157 at different concentrations (10^5^–10^7^ colony forming unit (CFU)/mL) (final conc.). As mentioned in Materials and Methods section, O157 at different CFUs (10^5^–10^7^ CFU) was loaded per module, and washed after infecting the gut cells for 4 h. During the 4 h, O157 grew, and some O157 detached from the gut cell layer during the washing. By visualizing the attached bacteria with O157 labeled with green fluorescent protein (GFP), it was found that the number of remaining O157 was similar to the number of loaded O157 (data not shown).

At 10^5^ CFU/mL, when O157 was lysed by CIP, the amount of released Stx2 was approximately 51.51 nM, and when lysed by GEN, the amount of Stx2 was less than 3 nM. This was quantified from the thickness of their bands in the Western blot (Appendix A). These concentrations of Stx2 did not affect the viability of the gut and kidney modules (Figure 4a). To investigate the effect of antibiotics treatment to O157 infection on the viability of Caco-2 and HKC-8 cells, LIVE/DEAD staining was performed. However, at 10^6^ and 10^7^ CFU/mL, many cells in both modules were killed or detached from the surface, relative to the cells in the gut and kidney modules without O157 infection (Figure 5a,b). These cytotoxic effects were more evident at 10^7^ CFU/mL (Figure 5b,d) than at 10^6^ CFU/mL (Figure 5a,c). The low viability of HKC-8 cells following O157 infection was consistent with their low TEER values (Figure 5e). If antibiotics were not administered after O157 infection, O157 would grow well for an additional 3 days. Thus, O157 could not only make the cell culture medium for Caco-2 cells acidic, but also secrete virulent factors [32]. This might be the reason why most of Caco-2 cells appeared to be dead (Figure 5a,b). The acidification of the culture medium by O157 occurred because the amount of culture medium in the module was limited, and was not periodically replaced. The low viability of HKC-8 cells could be due to Stx2 and other toxins, including hemolysin secreted from O157 in the gut module, because the bacteria could not cross the porous membrane (0.4 μm pore) in the gut module (Figure 2a). This is consistent with the results of a clinical study in which HUS occurred even in patients who did not receive antibiotic treatment after STEC infection [33]. These results suggest that an appropriate antibiotic therapy is necessary to protect the gut and kidney cells from O157.

To evaluate the impacts of antibiotic treatment of the gut compartment on the intoxication of cells in the kidney compartment, only the gut module previously seeded with either O157 10^6^ or 10^7^ CFU/mL was treated with either CIP or GEN at their respective MICs (0.06 and 15 µg/mL for CIP and GEN, respectively, (Appendix A)), and the kidney module was not treated with any antibiotics. The effect of each treatment on the viability of Caco-2 and HKC-8 cells was investigated using LIVE/DEAD staining 72 h after treatment (Figure 5a,b). It was confirmed that all bacteria were lysed within 24 h after each antibiotic treatment through colony counting. Irrespective of the bacterial concentration, the viability of Caco-2 cells treated with CIP was significantly higher than that of untreated Caco-2 cells (*p* < 0.01, 10^6^ CFU/mL; *p* < 0.001, 10^7^ CFU/mL) (Figure 5c,d). However, the effect of CIP treatment on the viability of HKC-8 cells depended on the bacterial concentration. At 10^6^ CFU/mL, the viability of HKC-8 cells was significantly improved by the antibiotic treatment compared to that of untreated HKC-8 cells (*p* < 0.01) (Figure 5a,c). At 10^7^ CFU/mL, there was no significant difference between the treated and untreated HKC-8 cells (*p* > 0.05) (Figure 5b,d). The low viability from the quantification of LIVE/DEAD staining could be due to the increase in Stx2 expression by CIP treatment, as has been shown by real-time quantitative polymerase chain reaction PCR (RT-qPCR) and Western blot (Appendix A), which highly depends on bacterial concentrations (Appendix A) [34,35]. It is also possible that Stx-containing outer membrane vesicles (OMVs) from O157, whose size is less than 200 nm in diameter, exerted cytotoxic effects on the kidney cells; moreover, they could be delivered through the perfusion, as since it has been reported that CIP can upregulate OMV-associated Stx2a [36,37].

Overall, the GEN treatment improved the viability of HKC-8 cells in the chip. There was no significant difference in the viability of Caco-2 cells between CIP and GEN treatments (Figure 5a–d). In contrast, there was a significant difference in the viability of HKC-8 cells between the two treatments (*p* < 0.05 for 10^6^ O157 CFU/mL; *p* < 0.01 for 10^7^ O157 CFU/mL) (Figure 5a–d). This is supported by the fact that the TEER value of CIP-treated HKC-8 cells was significantly lower than that of the GEN-treated HKC-8 cells. The bacterial lysates prepared by CIP treatment caused greater damage to the kidney cells than those prepared by GEN treatment (Appendix A). Similar results in Transwell plates have been reported elsewhere [34,35].

Our observations suggest that the amount of Stx produced by O157 varies depending on the antibiotic treatment, which can affect the development of HUS. Consequently, using GKA on chip, it was shown that the viability of kidney cells was affected by the concentration of O157 and type of antibiotics used to treat the O157 infection. CIP-induced toxin production from O157 and subsequent damage to kidney cells was significantly higher than that induced by GEN. These results are supported by animal studies showing that CIP treatment increases Stx2 expression in O157-infected mice, and in vitro studies showing the subsequent cytotoxicity of CIP-treated O157 to kidney cells [34,35]. To our knowledge, this is the first report on in vitro model systems capable of simulating the effects of antibiotic treatments on kidney damage in STEC-infected guts. This result suggests that the inappropriate treatment of STEC infection could exacerbate kidney damage as much as no treatment. This highlights the importance of the appropriate antibiotic treatment for STEC infections. GKA on chip may serve as a novel in vitro model system for screening the effects of antibiotic treatments, and if supported by further clinical studies, it may help provide a guideline for antibiotic treatments in cases of STEC infections.

In addition, GKA on chip can be used to observe the harmful effects of bacterial toxins on kidney cells without the complications of bacterial infection, because it is designed to prevent bacterial infection in the kidney module while maintaining bacterial infection in the gut module. This is possible because the membrane in the gut module prevents bacterial transmission to the kidney module.

One weakness of GKA on chip for the co-culturing of gut and kidney cells is the absence of vascular endothelium and immune cells. It has been reported that Stx elicits various inflammatory responses in the kidney endothelium, including cytokine production and the expression of P-selectin, thereby attracting leukocytes to the glomerulus [38,39]. The inclusion of additional components, such as blood vessels or immune cells, could improve the physiological relevance of GKA on chip and its predictive abilities as a disease model [40,41,42].

## 3. Conclusions

In this study, we developed GKA on a chip capable of co-culturing gut and kidney cells with fluidic connections and reproducing the crosstalk between the two organs. The STEC infection of gut cells and treatment with certain antibiotics resulted in the induction of Stx in the gut and damage to the kidney. Collectively, our results were consistent with clinical reports on O157 patients, in that CIP treatment in STEC infection aggravates kidney damage more than GEN treatment [43].

## 4. Materials and Methods

### 4.1. Design and Fabrication of GKA on Chip

The dimensions of the chip were 7.5 cm (L) × 2.5 cm (W) × 1.1 cm (H). It was fabricated by binding four layers: the top layer (6 mm thick) was made of polycarbonate (PC) using computer numerical control (CNC) machining; two middle layers with a thickness of 2 mm were made of polydimethylsiloxane (PDMS) (Sylgard^®^184) from Dow Corp. (Midland, MI, USA) by soft lithography [38]; and the bottom layer was a glass slide (Corning, Cortland, NY, USA). The molds for the PDMS layers were made of duralumin using CNC machining by Woosung Eng. (Seoul, Korea), respectively. The first mold was designed to have patterns for the two holes and two reservoirs, whereas the second mold was designed to have the same patterns on the first mold, but with an additional pattern of a channel connecting all of the modules and reservoirs (Appendix A).

To transfer the pattern from a mold to the PDMS layer, a mixture of a PDMS prepolymer and its curing agent at a ratio of 10 to 1 (*w*/*w*) was poured into each mold and cured at 80 °C in an oven for 2 h [44]. The PDMS layer was then peeled off from the mold. The module and reservoir patterns in the layer were opened using a biopsy punch and blade. The second PDMS layer and glass slide (1 mm) were treated with O_2_ plasma using a plasma machine (Femtoscience; Hwaseong, Korea) and were bound to each other while facing the channel side of the PDMS layer to the glass slide. The PC layer was pasted with the PDMS mixture using CleanTips swabs TX743B (TexWipe; Kernersville, NC, USA), and bound with the first PDMS layer with alignment in the modules and holes of both layers. They were then cured at 80 °C for 2 h to form a monolithic structure. A square cut (9 mm × 9 mm) of a polyester membrane (0.4 μm pore) from a 24 mm Transwell^®^ insert (Corning Inc.; Corning, NY, USA) was placed between the small holes (8 mm in diameter) of the PC/PDMS and PDMS/glass layers previously treated with the oxygen plasma.

The module for kidney cells was created by inserting a cut of polyester membrane between the two PC rings (inner and outer diameters: 8 and 16 mm, respectively), and fastening it with screws. For sanitization, the chip was autoclaved at 121 °C for 15 min. Ethanol (70%) was injected into the chip through a reservoir, and was rinsed three times with phosphate-buffered saline (PBS, pH 7.4; Gibco; Grand Island, NY, USA) using a pipette.

### 4.2. Determining the Concentration of Antibiotics

CIP and GEN were obtained from Sigma-Aldrich (St. Louis, MO, USA) were chosen as representatives of fluoroquinolone and aminoglycoside antibiotics. The MIC of each antibiotic was determined using the broth dilution method [45]. The MIC of CIP was 0.06 µg/mL, and that of the GEN was 15 µg/mL.

### 4.3. Cell Culture

The human epithelial colorectal adenocarcinoma cell line Caco-2 was obtained from American Type Culture Collection (ATCC; Manassas, VA, USA). The human renal proximal tubular cell line HKC-8 [46] was obtained from Seoul National University Bundang Hospital. Caco-2 cells were cultured in Dulbecco’s Modified Eagle’s Medium (DMEM)/High glucose (HyClone Laboratories; Logan, UT, USA) supplemented with 10% (*v*/*v*) fetal bovine serum (FBS) (HyClone Laboratories), 100 units/mL of penicillin (HyClone Laboratories), and 100 μg/mL of streptomycin (HyClone Laboratories), whileHKC-8 cells were cultured in DMEM/F12 (Gibco) supplemented with 5% (*v*/*v*) FBS, 100 units/mL of penicillin (Sigma-Aldrich) and 100 μg/mL of streptomycin (Sigma-Aldrich). The cells were maintained at 37 °C with 5% CO_2_. Cell culture medium was changed every 2 days.

### 4.4. Culture of Gut and Kidney Cells on the Chip

About a total of 6 × 10^4^ Caco-2 cells in 150 μL of DMEM/High glucose were seeded in the gut module on the chip using a pipette (Figure 2b). They were perfused with media in the reservoirs for 7 days by a gravity-driven flow using a tilting mechanical stage (Thorlabs; Newton, NJ, USA) with a custom-built computer program Kinesis^®^ (Thorlabs) at 37 °C with 5% CO_2_. This generated continuous perfusion between the gut and kidney chambers by the gravity-induced flow when the chip on the custom-made swing machine (Thorlabs) (Figure 2d) were periodically tilted at 10 degrees (0.1 degree/s) every 10 min (Figure 2c), as described in our previous reports [16,47].

About 2 × 10^4^ HKC-8 cells in 200 μL of DMEM/F12 were seeded in the kidney module and cultured in six well plates for 3 days at 37 °C with 5% CO_2_. The module was then inserted into the chip using tweezers, and Caco-2 cells were cultured for 7 days (Figure 2b). Finally, DMEM/F12 supplemented with 5% (*v*/*v*) FBS, 100 units/mL of penicillin, and 100 μg/mL of streptomycin was perfused into the chip using the tilting mechanical stage at 37 °C with 5% CO_2_ for 7 days.

### 4.5. Treatment of Stx2 on the Chip

Stx2 (List labs; Campbell, CA, USA) was diluted in the DMEM/F12 as supplemented with 5% (*v*/*v*) FBS, 100 unit/mL penicillin, and 100 μg/mL streptomycin. One hundred microliters of Stx2 was added only in the gut module, i.e., after assembly of the gut and kidney modules.

### 4.6. Bacterial Culture

An O157 strain (*stx1*^+^, *stx2*^+^) (43894) was purchased from ATCC. A single colony of O157 obtained from a Luria-Bertani (LB) (Sigma-Aldrich) agar plate was inoculated into 5 mL of LB broth and incubated at 200 rpm and 37 °C overnight. The next day, 50 μL of the culture was inoculated in 5 mL of fresh LB broth and incubated at 200 rpm and 37 °C for 2 h.

### 4.7. O157 Infection and Co-Culture

One milliliter of fresh bacterial culture was centrifuged at 13,000 rpm for 10 min, and the supernatant was removed. Bacterial pellet was resuspended in DMEM/F12 without the FBS and antibiotics to make its OD_600_ to be 1. The resuspended bacteria were diluted with DMEM/F12 to obtain O157 solutions at different concentrations (10^6^–10^8^ CFU/mL). Each 100 µL of the solution was added to the gut module, resulting in a multiplicity of infection (MOI) of about 10 to 1000; this was calculated to be equivalent to 10^5^ to 10^7^ CFU/mL. The solution was incubated at 37 °C in a CO_2_ incubator under static Caco-2 cells to be infected [38]. After 4 h, the culture medium on the gut module was removed, and Caco-2 cells were washed three times with 200 µL of PBS.

Next, 200 µL of DMEM/F12 as supplemented with 5% FBS and either CIP or GEN at their MIC was added to Caco-2 cells. The module in which HKC-8 cells had been cultured was assembled into the chip, as described above. Finally, the chip was incubated for 3 days on the tilting mechanical stage to generate the gravity-driven flow, allowing Caco-2 and HKC-8 cells to be perfused with the culture medium.

### 4.8. Cytotoxicity Test

Live and dead cells were fluorescently visualized using LIVE/DEAD™ Viability/Cytotoxicity Kit (Life Technologies, Eugene, OR, USA) as suggested by the manufacturer’s instructions. Fluorescent images of stained cells were acquired using a DeltaVision Elite fluorescence microscope (GE Healthcare, Chicago, IL, USA). The green and red fluorescence intensities were measured using ImageJ (NIH, USA), and the cell viability was calculated as the ratio of the red to green fluorescence intensities. The relative viability was calculated as the number of viable cells divided by the number of viable cells in the control from LIVE/DEAD-stained images.

### 4.9. Immunostaining

Caco-2 cells in the chip were carefully washed with PBS, fixed using 4% paraformaldehyde (Sigma-Aldrich) at 37 °C for 10 min, and permeabilized with 0.1% Triton X-100 (Sigma-Aldrich) at room temperature for 15 min. To reduce non-specific binding, the cells were blocked with PBS containing 2% bovine serum albumin (Gibco). Cells were incubated with mouse anti-occludin antibody (Invitrogen; Carlsbad, CA, USA; #33-1500) at 4 °C overnight, incubated with anti-mouse 488 antibody (Abcam; ab150113) at room temperature for 45 min, and stained with 4,6-diamidino-2-phenylindole (DAPI) from Thermo Fischer Scientific (Waltham, MA, USA).

### 4.10. Transepithelial Electrical Resistance (TEER) Measurement

TEER values were measured using an STX01 adjustable chopstick electrode and EVOM2 Volt/Ohm meter (World Precision Instruments, Sarasota, FL, USA). To facilitate the measurement, a 3 mm-thick PDMS layer with a hole (8 mm) was used as a guide (Appendix A). Sterilization was performed according to the manufacturer’s instructions. First, the electrode was immersed in 70% ethanol for 15 min. After drying in air for 15 s, it was rinsed with a sterilized 0.1 M NaCl (Sigma-Aldrich) solution and used for the TEER measurement. By measuring the resistance value of the module, the TEER value of the cell layer was obtained using Equation (1), as follows:(1)$$\mathrm{TEER}=\left(R_{total}-R_{blank}\right)\times A.$$

In the above, *R_total_* is the resistance value of the module with cells, *R_blank_* is the resistance value of the module without cells, and *A* is the area of the space in which the cells are cultured.

### 4.11. Numerical Simulation of Stx2 Transport in the Chip

To gain a better understanding of the kinetics of the transport and distribution of Stx inside the chip, we developed a simplified mathematical model for Stx2 transport in the chip. The model was constructed as a two-module model (gut and kidney) connected via a blood vessel. It was assumed that the transport of Stx2 across the gut epithelium is a rate-limiting step (diffusion-limited), i.e., perfusion along the blood vessel does not present a rate-limiting step. This was based on the observation that the gut epithelial cells do not express a Stx2 receptor, Gb3; this can facilitate endocytosis of Stx, and the major route of Stx2 transport across the gut epithelium is paracellular transport [29,48]. On the other hand, Gb3 is known to be expressed in the endothelium, and Stx can readily enter the Gb3-positive cells through Gb3, which explains the sensitivity of the kidney tissue to exposure to even a low concentration of Stx [49]. Hence, we assumed that Stx2 could penetrate relatively freely across the endothelial barrier, and that translocation across the gut barrier was the major rate-limiting step. Although it is highly probable that the transport of Stx2 across the kidney endothelium occurs faster than the transport across the gut epithelium, we could not find relevant quantitative data for Stx2 transport across the kidney endothelium; therefore, we used the same values for both directions.

The mass balance Equations (2) and (3) for the two-module model were set up as follows:(2)$$\frac{dC_{1}}{dt}=\frac{PA}{V_{1}}\left(C_{1}-C_{2}\right)$$
(3)$$\frac{dC_{2}}{dt}=\frac{PA}{V_{2}}\left(C_{2}-C_{1}\right)$$

Here, *C*_1_ (ng/cm^3^) is the concentration in the gut module, *C*_2_ (ng/cm^3^) is the concentration in the kidney module, *V*_1_ (cm^3^) is the gut module volume, *V*_2_ (cm^3^) is the kidney module volume, *t* (s) is the time, *P* (cm/s) is the permeability constant, and *A* (cm^2^) is the cell culture area.

The permeability constant was determined based on a previous study that measured the amount of Stx2 transported across the gut epithelium at a given time [25]. Based on this measurement, we calculated the apparent permeability constant for the gut epithelium using Equation (4).
(4)$$P=\frac{dQ}{dt}\times \frac{1}{C_{0}A},$$

In the above, *P* (cm/s) is the permeability constant, *Q* (ng) is the amount of Stx2 transported, *C*_0_ (ng/cm^3^) is the initial apical concentration, *A* (cm^2^) is the cell culture area, and *t* (s) is the time. *Q*, *C*_0_, *A*, and *t* were determined by the methods and results described in the reference, and *P* was calculated as 2.1 × 10^−8^ cm/s.

### 4.12. Statistical Analysis

All data were presented as the mean ± standard deviation. Comparisons of the mean values between the two groups were performed using Student’s *t*-tests. The levels of statistical significance were set at * *p* < 0.05, ** *p* < 0.01, and *** *p* < 0.001.

## Figures and Tables

**Figure 1 toxins-13-00775-f001:**
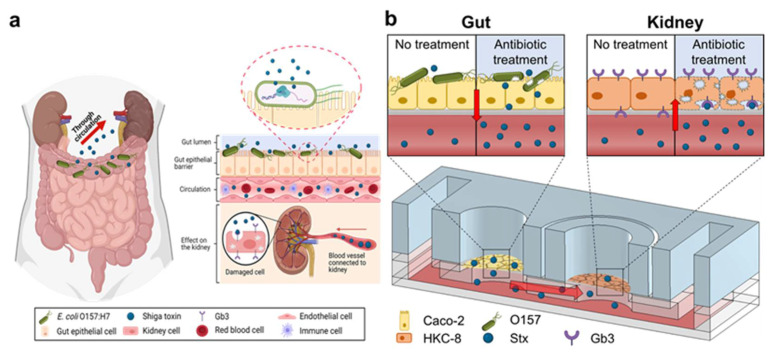
Design of gut–kidney axis (GKA) on chip for co-culture of gut (Caco-2) and kidney (HKC-8) cells. (**a**) The anatomy of the gut infected by O157 and the effect of Shiga toxin 2 (Stx2) on the kidney. (**b**) Schematic showing hemolytic–uremic syndrome (HUS) development in kidney cells by Stx released by O157 in gut cells in the chip during the antibiotic treatment. The image in (**a**) was created with BioRender.com.

**Figure 2 toxins-13-00775-f002:**
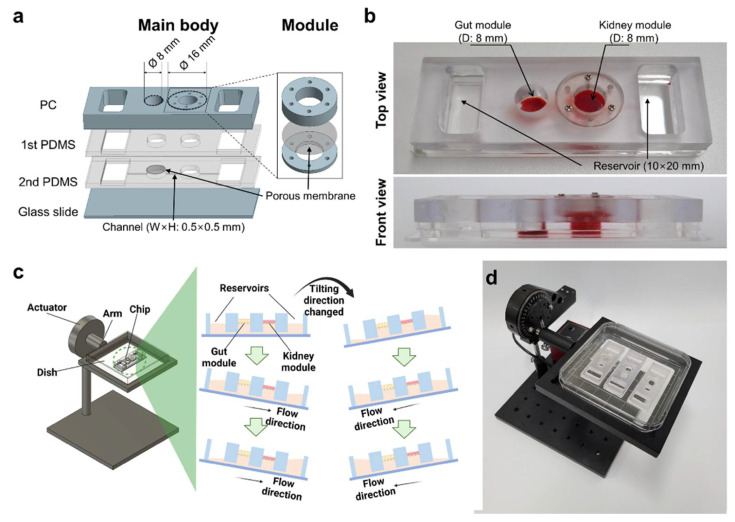
Fabrication and operation of GKA on chip for co-culturing of gut and kidney cells. (**a**) Assembly of a main body in four layers with modules. (**b**) View of completed GKA on chip showing gut and kidney modules and two reservoirs. (**c**) Gravity-induced perfusion by periodically tilting the chip 10 degrees (0.1 degree/s) every 10 min. (**d**) View of the tilting machine inducing gravity-driven perfusion of cell culture medium in GKA on chip. The image in (**c**) was created with BioRender.com.

**Figure 3 toxins-13-00775-f003:**
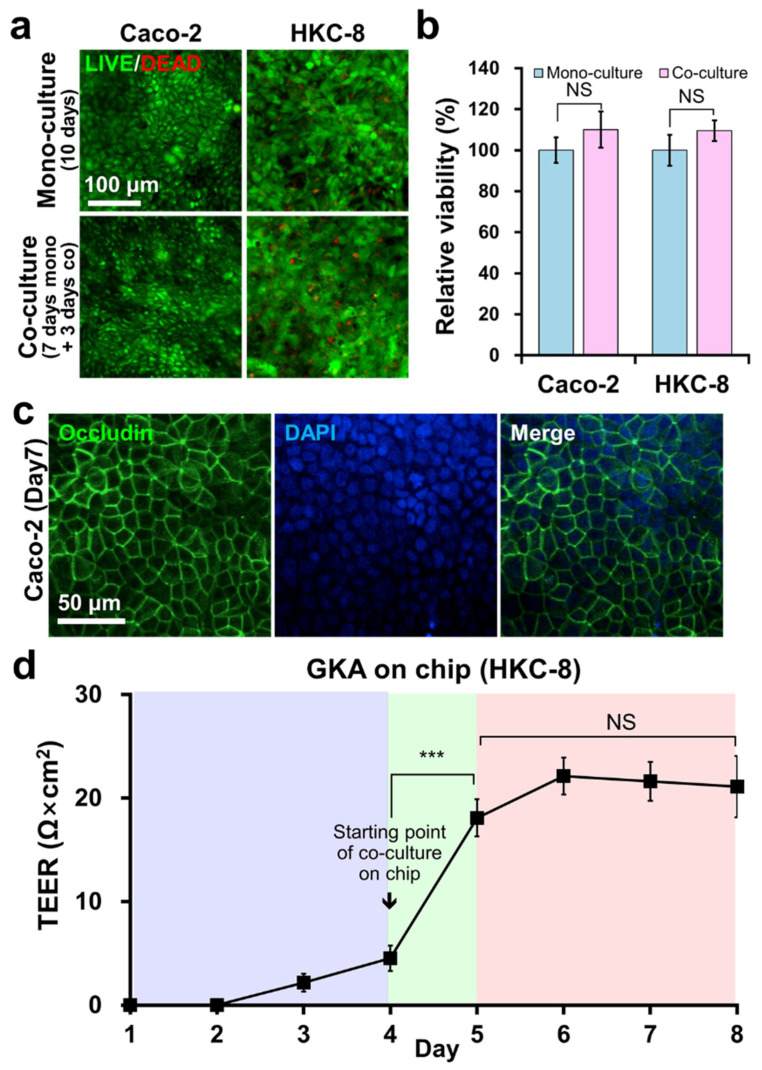
Viability and junctional integrity of gut and kidney cells on the chip. (**a**) LIVE/DEAD stained images and (**b**) EZ-CytoX assay of Caco-2 and HKC-8 cells either mono-cultured or co-cultured for 3 days. (**c**) Immunostaining of occludin in Caco-2 cells at day 7. (**d**) Transepithelial electrical resistance (TEER) values measured daily over 8 days in HKC-8 cells mono-cultured in the kidney module for 4 days and then co-cultured in the chip for the remaining days of the measurement. Each experiment was repeated three times. Student’s *t*-test. NS; not significant. *** *p* < 0.001.

**Figure 4 toxins-13-00775-f004:**
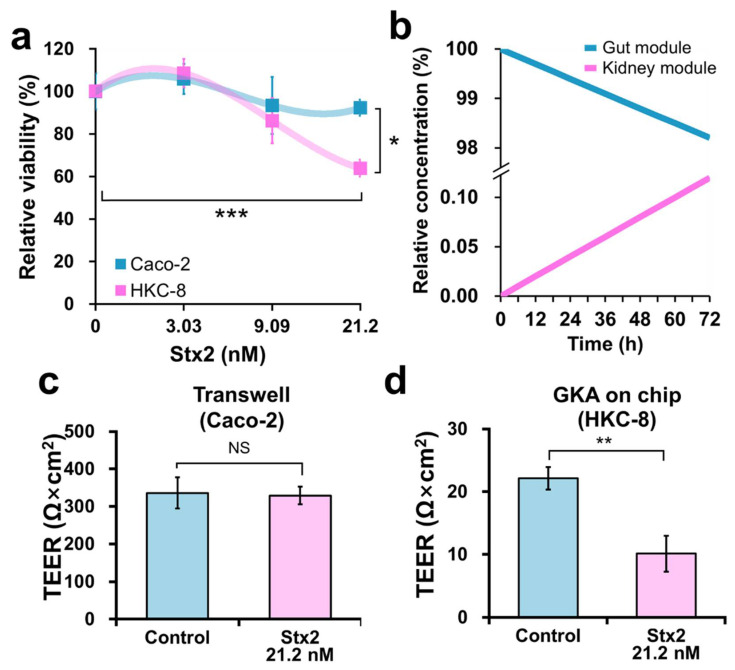
Effect of Stx2 on cellular integrity of gut and kidney cells in GKA on chip. (**a**) Viabilities of Caco-2 and HKC-8 cells to Stx2 in GKA on chip. Only the gut module was treated with the toxin at different concentrations (0–21.2 nM) for 72 h. (**b**) Simulation of Stx2 transport from the gut module to the kidney module using the Equations (2) and (3) in the Experimental Section. (**c**) TEER Value of Caco-2 cells to Stx2 at 21.2 nM in Transwell for 72 h. (**d**) TEER values of HKC-8 cells after treatment with 21.2 nM of Stx2 in the gut module for 72 h. Sample number (n) = 3, Student’s *t*-test. NS; not significant, * *p* < 0.05, ** *p* < 0.01, *** *p* < 0.001.

**Figure 5 toxins-13-00775-f005:**
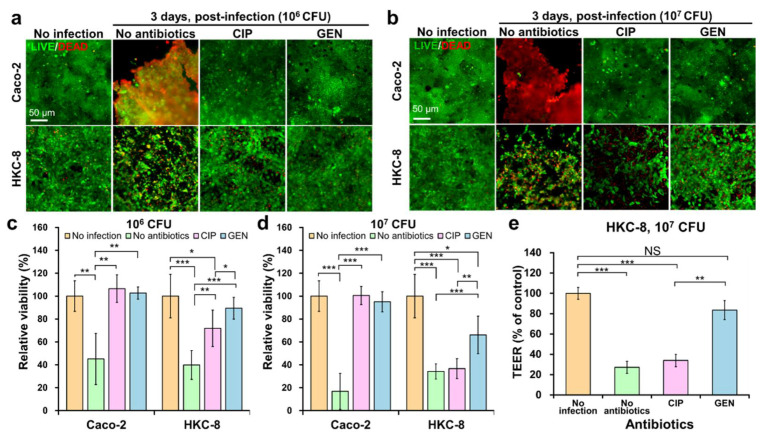
Effect of antibiotics treatment to O157 infection on the viability and integrity of gut and kidney cells in the chip. (**a**,**b**) LIVE/DEAD stained images from 10^6^ and 10^7^ CFU per module and (**c**,**d**) cell viability of Caco-2 and HKC-8 cells being treated with O157 lysed by either CIP or GEN in the chip for 72 h. The relative viability was calculated as the number of viable cells divided by the number of viable cells in the control (no infection and no antibiotics) from LIVE/DEAD stained images. (**e**) TEER values of HKC-8 cells in module of the chip when the gut module was infected with O157 at 10^7^ CFU and treated with either CIP or GEN for 72 h. n = 3, Student’s *t*-test, NS; not significant, * *p* < 0.05, ** *p* < 0.01, *** *p* < 0.001.

## Data Availability

Not applicable.

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
