# Peer review of "Gut–Kidney Axis on Chip for Studying Effects of Antibiotics on Risk of Hemolytic Uremic Syndrome by Shiga Toxin-Producing Escherichia coli"

_toxins, 2021, doi:10.3390/toxins13110775_

Round 1

Reviewer 1 Report

In the manuscript “Gut-Kidney-axis on chip to study effects of antibiotics on risk of HUS by STEC” the authors developed a chip to cultivate kidney and gut cell cultures and show first experiments how this gut-kidney-axis chip could be used to analyze the effect of STEC infections and parallel antibiotic treatment.

The overall idea of the study is interesting. However, the presentation of study can be majorly improved.

Major general comments on the manuscript are:

  • The common thread in the introduction is partially hard to follow and especially paragraph 1 and 2 should be revised.
  • As the construction of the chip is a huge part of the study itself, I would recommend to describe the chip design in the results and discussion part and move figure 1 to the this part. Please add more details on the chip design and experimental aspects, as are given so far in the introduction.
  • Add more detailed description on all results presented
  • Think about to separate the results from the discussion part.
  • Structure of the Results and Discussion part is not always clear and important information is hidden in figure captions, which should be addressed in the main text.
  • Control of integrity of cell cultures measured with TEER: What is the effect of cultivation on the cell cultures in different “wells”. At least one control is missing here, to show that measuring the TEER values in Transwell are comparable to the one measured in the modules.
  • Detection of cell viability not possible in the presence of O157. Would alternative Assays or staining techniques allow to gain results here?
  • Results for the MIC detection not presents
  • Instead of using a simulation is there a possibility to detect the amount of Stx in the culture supernatant experimentally?
  • Experimental procedure descriptions and description of results are partially contradictory
  • Experimental procedure does not clarify when and how Stx2 was used.

Detailed comments:

Line 30-33: explain in more detail

Line 36-38: explain in more detail

Line 93-95: unclear, please explain

Line110: figure 1 d: depicted line is this a reasonable fit of the TEER value or just used to illustrate the data point?

Line 118: Figure 3 a: some as in Line 110.

Line 127: treated? Correct word?

Line 143-145: discrepancy to what was stated before.

Line 179: include date in supplement

Line 211-213: how detected?

Line 326: overnight for 2 h à contradictory

Line 331: What about 10^5 CFU/mL?

Comments on the supplement:

Figure S1, S2, S3d: Same as for line 110.

Include Figure S3 in the main manuscript and discuss in greater detail.

Author Response

Reviewer #1

Comment 1. The common thread in the introduction is partially hard to follow and especially paragraph 1 and 2 should be revised.

Response: We are sorry for the inconvenience in following the text. We added explanations to make the manuscript easy to follow throughout the MS, especially in paragraphs 1 and 2.

Comment 2.  As the construction of the chip is a huge part of the study itself, I would recommend to describe the chip design in the results and discussion part and move figure 1 to the this part. Please add more details on the chip design and experimental aspects, as are given so far in the introduction.

Response: We appreciate this insightful suggestion. We separate the fabrication part (Figure 1c, d) from Figure 1 and make it as Figure 2 to answer your request. We added detailed explanation about chip design accordingly.

Comment 3. Add more detailed description on all results presented

Response: We tried to add details throughout the results.

Comment 4. Think about to separate the results from the discussion part.

Response: We appreciate the reviewer’s comment. However, we think the current combined format could be better to describe our results due to the complexity of the results. I hope you understand this.

Comment 5. Structure of the Results and Discussion part is not always clear and important information is hidden in figure captions, which should be addressed in the main text.

Response: We moved some details in the figure captions to the main text as requested.

Comment 6. Control of integrity of cell cultures measured with TEER: What is the effect of cultivation on the cell cultures in different “wells”. At least one control is missing here, to show that measuring the TEER values in Transwell are comparable to the one measured in the modules.

Response: We appreciate the reviewer’s comment. When over confluent, HKC-8 layer in the well quickly became very unstable and easily detached from the culture surface since they are fast-growing cells and metabolic wastes are accumulated. Therefore, it was quite challenging to measure TEER value of HKC-8 in the wells. However, the wastes are washed out when the kidney module is perfused in the chip, and the kidney cells were maintained better than in the well. That is why we could present only TEER of the chip.

Also, when HKC-8 cells were cultured in wells, reported TEER values were in the range of 9~11 ohms × cm2. When we measured TEER values of HKC-8 cells in the chip, the values were considerably higher than the reported TEER values in wells. Therefore, we could conclude that the HKC-8 cells formed tight barriers in the chip. We added a sentence in the section 2.2. of the Results and discussion to regard this difference in TEER values between our chip and the wells as below.

“TEER values in the kidney module increased until Day 5 and then became stable at 20 ohms × cm2 without significant changes throughout the following days (Figure 2d). The TEER values were higher than previously reported values, which indicates the kidney cells formed tight barriers in the chip.”

Comment 7. Detection of cell viability not possible in the presence of O157. Would alternative Assays or staining techniques allow to gain results here?

Response: It was our mistake to leave the sentence. Previously, we cannot detect the cell viability but, in this manuscript, we had no problem in the detection by using the method. Thus, we deleted the sentences about the impossibility of detecting the viability of Caco-2 cells and edit the method.

Comment 8. Results for the MIC detection not presents

Response: We added the picture of colony counting for determining MIC of each antibiotic into supplementary (Figure S3).

Comment 9. Instead of using a simulation is there a possibility to detect the amount of Stx in the culture supernatant experimentally?

Response: We quantified the amount of Stx in the culture supernatant by comparing the thickness of band from western blot. We quantified the amount of Stx2 from lysed O157 from Figure S4b and added its standard curve in Figure S4b and its approximated amount in the main text (Line 220).

Comment 10. Experimental procedure descriptions and description of results are partially contradictory

Response: We tried to revise the description of procedures and results to address the issue. The changes are found in our responses in detailed comments

Comment 11. Experimental procedure does not clarify when and how Stx2 was used.

Response: We added methods of when and how Stx2 was used at Subsection 4.5.

Detailed comment 1. Line 30-33: explain in more details.

Response: Thanks for the kind comment. We added more details about the relation between antibiotic treatment to STEC infection and mortality.

Detailed comment 2. Line 36-38: explain in more details.

Response: We added more details about the action of Stx2 in the body.

Detailed comment 3. Line 93-95: unclear, please explain

Response: Thanks for the kind comment. We added more details about advantage of our chip capable of co-culturing two types of cells.

Detailed comment 4. Line110: figure 1 d: depicted line is this a reasonable fit of the TEER value or just used to illustrate the data point?

Response: The line originally meant to be the trend line of the data, but we agree with the reviewer that it is not suitable in this graph. Therefore, we have modified the figure and drawn a line connecting the data points.

Detailed comment 5. Line 118: Figure 3 a: some as in Line 110.

Response: We have modified the figure and drawn lines connecting the data points.

Detailed comment 6. Line 127: treated? Correct word?

Response: We agree that our expression was awkward and think that ‘applied’ is better expression. Below is the modified sentence.

“The viability of Caco-2 and HKC-8 cells was investigated when purified Stx2 was applied at a concentration of 100 to 700 ng/mL in the gut module of the GKA on chip.”

Detailed comment 7. Line 143-145: discrepancy to what was stated before.

Response: We measured the TEER of Caco-2 and HKC-8 cells from Transwell and GKA on chip, respectively. We revised the sentence as below.

“To confirm the effect of Stx2 on the integrity of Caco-2 and HKC-8 cells, 21.2 nM of Stx2 was treated to the Caco-2 cells in Transwell and treated to the gut module of GKA on chip. TEER of Caco-2 cells and HKC-8 cells was measured 72 h after the treatment (Figure 4c, d).”

Detailed comment 8. Line 179: include date in supplement

Response: We used the approximate amount of Stx2 from O157 105 CFU to explain why we didn't include 105 CFU/mL. It was not sufficient to induce cytotoxicity in kidney cells.

Detailed comment 9. Line 211-213: how detected?

Response: We added explanation about how we detected the low viability and increase in Stx2 expression.

Detailed comment 10. Line 326: overnight for 2 h à contradictory

Response: We deleted the word ‘overnight’.

Detailed comment 11. Line 331: What about 10^5 CFU/mL?

Response: When 105 CFU/mL are used, 104 CFU of O157 were loaded into the chip. In this case, their infection in the gut module would not occur. That is why we did not include the concentration in the study.

Reviewer 2 Report

In this manuscript, the authors designed and developed a novel “gut-kidney-axis on chip” system which could co-culture two different cell lines as representatives of gut and kidney epithelia respectively and have been utilized to study EHEC infection with or without antibiotics treatment. Generally speaking, this is a very interesting story. The experiments were designed and performed well, the manuscript was also organized and written well. The newly developed system could potentially benefit the field a lot. To move the manuscript forward to publishable level, couple of minus issues need to be addressed:

  1. Line 35, Gb3’s full name is globotriaosylceramide but not “globotriaosylceramide-3”.
  2. As a critical control, a toxin negative strain (Stx1-/-, Stx2-/-) should be included in the co-culture experiments to further demonstrate the cytotoxicity is induced by toxins.

Author Response

Comment 1. Line 35, Gb3’s full name is globotriaosylceramide but not “globotriaosylceramide-3”.

Response: Thank you for mentioning the typo. We corrected it.

Comment 2. As a critical control, a toxin negative strain (Stx1-/-, Stx2-/-) should be included in the co-culture experiments to further demonstrate the cytotoxicity is induced by toxins.

Response: Thank you for the valuable comment. Some non-STEC strain sometimes caused HUS. Due to the time limitation of this revision process (5 days), we will consider your suggestion to be included in the future report. We hope you understand this.

Reviewer 3 Report

The authors describe a fluidic device with two compartments that allow to grow independently gut and kidney cells. Then both compartments are put together and connected with a fluidic channel.  The gut compartment can be infected with STEC bacteria. When Shigatoxin is released by antibiotic treatment, the kidney cells are subjected to toxicity. The authors show that various antibiotics can be studied, including their differential effects on Stx production and subsequent kidney cell intoxication.

The device seems clever and functional. It should help reduce animal experiments and provide data more relevant to the human disease as compared to animal models. The experiments seem carefully done with proper controls.

A limit could be the use of occluding labelling to assess epithelial integrity on the gut side, due to impossibility to perform electric measurements. The authors perform independent Caco-2 transwell measurements to overcome this issue. Another limit may be the absence of endothelial barrier and immune cells. This point is discussed in the article.

Comments:

Introduction:

The “German” epidemics of 2011 (that was European although it started in Germany) was due to the O104:H4 serotype, which is not mentioned in the introduction. The text may wrongly suggest it was due to the O157:H7 serotype and should be corrected.

It is mentioned that the MOOC can help predict PK-PD. However, this would necessarily imply the presence of liver cells that are crucial for metabolism. This part should be rephrased not to imply that the gut-kidney chip described in this work would allow PK-PD prediction.

Results

It is not clear how a tilting machine would perfuse the two cells through a thin channel. Could the authors clarify? An additional scheme or picture would help. Same with the infection procedure: the bacterial culture is removed, so it is inferred that some bacteria adhere to the Caco-2 layer? This means that much less than the 10(5) or 10(7) pfu remain in the Caco-2 well? Please give more details in the main text.

Figure 3: 700 ng/mL of Stx2 in the gut compartment leads to ~0.7 ng/mL in the kidney compartment. Please convert also to toxin molarity. The authors indicate that the effect of the toxin is most obvious, especially at 700 ng/mL, however, the curve show it is only visible at 700 ng/mL. Please correct.

Fig. 3c is confusing. It shows TEER values for gut cells in transwell but the main text does not state clearly that transwells were used in place of the gut compartment of the MOOC. More detail is given later in the text so this point is hard to follow. Please clarify the main text.

The authors should not draw conclusions about patient treatment within the Results and discussion part. The authors should discuss this point at the very last only. Although the gut kidney chip is a very interesting device that could greatly help to screen for antibiotics for potential prevention of HUS, it clearly doesn’t replace the complexity of the whole organism. Decisions to test new antibiotics for STEC infections may need additional arguments. The authors should properly tune their conclusions about this.

Page 7, second paragraph, the authors state that they “evaluate the risk of HUS by antibiotic treatment”. Instead, they should stick to the capacity of their system and mention in place: “evaluate the impact of antibiotic treatment of the gut compartment on the intoxication of cells in the kidney compartment”. Relevance to HUS should be addressed only in the Conclusion (or the very end of the Results and discussion). Also, in this same paragraph, the authors should make very clear if the antibiotics are applied only in the gut compartment, which seems to be the case but is not so clear.

The sentence line 213-215 is confusing and must be clarified.

Line 257: please site the clinical studies. One study? Several studies?

Stx are secreted or produced by O157 in free soluble form, but also associated with outer membrane vesicles. Did the authors try to analyze in which form is the toxin reaching the kidney compartment? Could these outer membrane vesicles cross the filters in both chambers? Please add comments in the discussion.

Check spelling for “mould”. Mold?

Figure S3a : The ordinate axis is expressed as fold increase. Should we understand that “No treatment” equals 1? Please specify in the Figure legend. Also, the legend of the axis and of the Figure text should be corrected to stx2 (italics, the gene) instead of Stx2 (the toxin) or stx2.

Are the authors sure that the bacteria are lysed by CIP or GEN? Isn’t there toxin secretion together with some lysis? Please correct adequately. The legend of the Western in 3b should clearly indicate if Stx2A is indeed the A chain and indicate whether this is an SDS-PAGE under reducing conditions. Legend S3d: how many wells for well plate?

The first Figure S5: should be labeled S4; mold instead of mould?

The whole text needs English native speaker check and correction.

Author Response

Comment 1. The “German” epidemics of 2011 (that was European although it started in Germany) was due to the O104:H4 serotype, which is not mentioned in the introduction. The text may wrongly suggest it was due to the O157:H7 serotype and should be corrected.

Response: Thanks for pointing out our mistake. We’ve replaced the examples with outbreaks by O157:H7 serotype.

Comment 2. It is mentioned that the MOOC can help predict PK-PD. However, this would necessarily imply the presence of liver cells that are crucial for metabolism. This part should be rephrased not to imply that the gut-kidney chip described in this work would allow PK-PD prediction.

Response: We appreciate the reviewer’s comment. We agree that 2nd paragraph in the introduction may let readers misunderstand our chip would allow PK-PD prediction. Below is the added sentence in 2nd paragraph of the introduction to make the sentence clearer.

“The complex process that drugs undergo inside the body, including the absorption, distribution, metabolism and excretion (ADME), involves different organs in the body, and earlier proof-of-concept development of MOOC for predicting pharmacokinetics-pharmacodynamics of drugs has been reported [16-18]. Therefore, many current MOOCs have made it possible to elucidate disease mechanisms and develop therapeutic strategies for diseases [18].”

Comment 3. It is not clear how a tilting machine would perfuse the two cells through a thin channel. Could the authors clarify? An additional scheme or picture would help. Same with the infection procedure: the bacterial culture is removed, so it is inferred that some bacteria adhere to the Caco-2 layer? This means that much less than the 10(5) or 10(7) pfu remain in the Caco-2 well? Please give more details in the main text.

Response: We appreciate the reviewer’s recommendation. The perfusion in the chip was enabled by gravity-induced flow, by tilting the chip using computer-controlled tilting device. When the chip on the machine is tilted, hydraulic head is created in the chip. This pressure is enough to generate fluid flow through the thin channel, and the medium will be eventually collected in the lower reservoir of the chip. The machine tilts the chip periodically. Then, the flow is continuously generated through the channel. The schematic on this is added in the revised manuscript as Figure 2c. Plus, we’ve added some explanation about the number of remaining bacteria in the well after infecting and washing as below.

“As mentioned in Materials and Methods sections, O157 105-107 CFU was loaded per module and washed after infecting gut cells for 4 hours. During the 4 hours O157 doubled and during washing some O157 detached from the gut cell layer.” (Line 210)

Comment 4. Figure 3: 700 ng/mL of Stx2 in the gut compartment leads to ~0.7 ng/mL in the kidney compartment. Please convert also to toxin molarity. The authors indicate that the effect of the toxin is most obvious, especially at 700 ng/mL, however, the curve show it is only visible at 700 ng/mL. Please correct.

Response: We appreciate the kind comment. We’ve converted the units of Stx2 from weight per volume to toxin molarity. We’ve revised the sentence as below.

“The inhibitory effect of Stx2 on HKC-8 cells was obvious at 21.2 nM (P < 0.001) (Figure 4a).” (Line 162)

Comment 5. Fig. 3c is confusing. It shows TEER values for gut cells in transwell but the main text does not state clearly that transwells were used in place of the gut compartment of the MOOC. More detail is given later in the text so this point is hard to follow. Please clarify the main text.

Response: Thank you for your suggestion. We revised and placed the text explaining why we used Transwell for measuring TEER values of gut cells as below.

“It is difficult to measure TEER value in the microfluidic device [26] because the TEER electrodes are designed for Transwell and are not fit for the chip. Since the gut module could not be separated from the chip (Figure 2a), we could obtain TEER value from the kidney module only. To indirectly estimate the effect of Stx2 on the integrity of Caco-2 cells, Caco-2 cells were cultured in Transwell under the same conditions as the chip and their TEER was measured after Stx2 treatment.” (Line 177)

Comment 6. The authors should not draw conclusions about patient treatment within the Results and discussion part. The authors should discuss this point at the very last only. Although the gut kidney chip is a very interesting device that could greatly help to screen for antibiotics for potential prevention of HUS, it clearly doesn’t replace the complexity of the whole organism. Decisions to test new antibiotics for STEC infections may need additional arguments. The authors should properly tune their conclusions about this.

Response: Thank you for your insightful comments. We moved sentences about testing new antibiotics for STEC to Conclusion and tune downed the tone of that sentences.

“Collectively, our results were consistent with clinical reports in O157 patients in that CIP treatment in STEC infection aggravates the kidney damages more than GEN [43]. This result may suggest inappropriate treatment to STEC infection could exacerbate kidney damage as much as no treatment.” (Line 293)

Comment 7. Page 7, second paragraph, the authors state that they “evaluate the risk of HUS by antibiotic treatment”. Instead, they should stick to the capacity of their system and mention in place: “evaluate the impact of antibiotic treatment of the gut compartment on the intoxication of cells in the kidney compartment”. Relevance to HUS should be addressed only in the Conclusion (or the very end of the Results and discussion). Also, in this same paragraph, the authors should make very clear if the antibiotics are applied only in the gut compartment, which seems to be the case but is not so clear.

Response: Thank you for this thought-provoking suggestion. We revised the sentence as below.

“To evaluate the impact of antibiotic treatment of the gut compartment on the intoxication of cells in the kidney compartment” (Line 238)

Comment 8. The sentence line 213-215 is confusing and must be clarified.

Response: We appreciate the reviewer’s comment and agree that the mentioned part is confusing. We’ve divided the explanation about animal study and in vitro study so revised that sentence as below.

“These results are supported by animal studies showing that CIP treatment increased Stx2 expression in the O157-infected mice and in vitro studies showing that subsequent cytotoxicity of CIP treated O157 to kidney cells [34, 35].” (Line 273)

Comment 9.Line 257: please site the clinical studies. One study? Several studies?

Response: We added a reference of clinical studies about antibiotic treatment of STEC infection in Conclusion as below.

“Collectively, our results were consistent with clinical reports in O157 patients in that CIP treatment in STEC infection aggravates the kidney damages more than GEN [43].” (Line 293)

Comment 10. Stx are secreted or produced by O157 in free soluble form, but also associated with outer membrane vesicles. Did the authors try to analyze in which form is the toxin reaching the kidney compartment? Could these outer membrane vesicles cross the filters in both chambers? Please add comments in the discussion.

Response: We appreciate the insightful comment. To the extent of our knowledge, the size of Gram-negative bacterial OMVs can range from 20–250 nm. In case of O157, their OMV size seems less than 200 nm based on the DLS analysis conducted by Bielaszewska et al. in 2017 (PLoS Pathog 13(2): e1006159). As mentioned in the Materials & Methods, we used porous membrane (filter) with pore size of 0.4 micrometer for cell culture on the chip, so it is possible that O157 OMVs could cross the membrane. We have added this comment in the revised manuscript as follows in section 2.3.

“The low viability from quantification of LIVE/DEAD staining could be due to the increase in Stx2 expression by CIP treatment as shown by Real-time quantitative PCR (RT-qPCR) and western blot (Figure S4a, b), which highly depends on bacterial concentrations (Figure S4c, d) [34, 35]. Also, it is possible that Stx-containing outer membrane vesicles (OMVs) from O157, whose size is less than 200 nm in diameter, exerted cytotoxic effects on the kidney cells, delivered through the perfusion since it has been reported CIP can upregulate OMV-associated Stx2a [36, 37].”

Comment 11. Check spelling for “mould”. Mold?

Response: We modified the mistakes.

Comment 12. Figure S3a: The ordinate axis is expressed as fold increase. Should we understand that “No treatment” equals 1? Please specify in the Figure legend. Also, the legend of the axis and of the Figure text should be corrected to stx2 (italics, the gene) instead of Stx2 (the toxin) or stx2.

Response: We appreciate the reviewer’s kind comment. We added the explanation about normalizing data and calculating fold increase. We modified the legend of the axis and of the caption of Figure S3.

Comment 13. Are the authors sure that the bacteria are lysed by CIP or GEN? Isn’t there toxin secretion together with some lysis? Please correct adequately. The legend of the Western in 3b should clearly indicate if Stx2A is indeed the A chain and indicate whether this is an SDS-PAGE under reducing conditions. Legend S3d: how many wells for well plate?

Response: We appreciate reviewer’s comment. We are sure that the bacteria are lysed by CIP or GEN, because we observed no colonies were formed from the medium. We added this explanation in the manuscript. According to the company selling Stx2, band in SDS-PAGE around 33 kDa would be the A polypeptides. We used Laemmli sample buffer for preparation of samples for SDS-PAGE and 2-mercaptoethanol was added to the buffer as reducing agent. We added the number of wells for well plate in the legend S3d.

Comment 14. The first Figure S5: should be labeled S4; mold instead of mould?

Response: We corrected the mistake.

Comment 15. The whole text needs English native speaker check and correction.

Response: We appreciated the reviewer’s recommendation. We performed check and correction through a professional English editing company before submitting the first submission.